# Changes in Melatonin Concentration in a Clinical Observation Study under the Influence of Low-Frequency Magnetic Fields (Magnetic Stimulation in Men with Low Back Pain)—Results of Changes in an Eight-Point Circadian Profile

**DOI:** 10.3390/ijms242115860

**Published:** 2023-11-01

**Authors:** Marta Woldańska-Okońska, Anna Kubsik-Gidlewska, Kamil Koszela

**Affiliations:** 1Department of Internal Medicine, Rehabilitation and Physical Medicine, Medical University, 90-419 Lodz, Poland; marta.okonska@poczta.onet.pl (M.W.-O.); anna.kubsik@umed.lodz.pl (A.K.-G.); 2Department of Neuroorthopedics and Neurology Clinic and Polyclinic, National Institute of Geriatrics, Rheumatology and Rehabilitation, 02-637 Warsaw, Poland

**Keywords:** melatonin, circadian, magnetic field, physical therapy

## Abstract

The aim of this study was to assess the changes in melatonin concentration under the influence of magnetic stimulation in men with low back pain. A total of 15 men were used in this study, divided into two groups. In Group 1, consisting of seven men, the M1P1 Viofor JPS program was used twice a day for 8 min, at 08:00 and 13:00. In Group 2, consisting of eight men, the M2P2 Viofor JPS program was used once a day for 12 min at 10:00. The application was subjected to the whole body of patients. The treatments in both groups lasted 3 weeks, for 5 days each week, with breaks on weekends. The diurnal melatonin profile was determined the day before exposure and the day after the last treatment, as well as at one-month follow-up. Blood samples were collected eight times a day. In both programs, magnetic stimulation did not reduce the nocturnal peak of melatonin concentration. After exposure, prolonged secretion of melatonin was observed until the morning hours. The impact of the magnetic field was maintained 1 month after the end of the application. The effect of the magnetic field was maintained for 1 month from the end of the application, which confirms the thesis about the occurrence of the phenomenon of biological hysteresis. The parameters of the magnetic fields, the application system, and the time and length of the application may affect the secretion of melatonin.

## 1. Introduction

The basic physiological function of melatonin is to transfer information about the diurnal cycle of light and darkness, regulate circadian rhythms, and synchronize rhythms. Melatonin acts through its receptors, which are widely distributed in various tissues. Exposure to light activates the suprachiasmatic nucleus of the hypothalamus and inhibits melatonin production. Light can suppress or synchronize melatonin production, and melatonin secretion from the pineal gland, which promotes sleep, is strictly related to the period of darkness. If the rhythmic secretion of melatonin is disrupted, the circadian rhythms of sleep and wakefulness are also disrupted, known as circadian rhythm sleep-wake disorder (CRSWD) [1,2].

At the molecular level, circadian rhythms are controlled by specific ‘clock genes’. These genes and their protein products are synchronized with a 24 h circadian period based on inputs from external and internal timing signals and affect the cycles of circadian rhythms such as sleep, wake cycles, and cycles of body temperature, blood pressure, and metabolism [3]. 

Studies of shift workers assessed the risk of various types of cancer associated with disturbances in the circadian rhythm of melatonin release as high (especially prostate and colorectal cancer and other endocrine cancers, including ovarian and endometrial cancer), as well as the risk of non-cancer effects such as cardiovascular disease and diabetes [3].

The Society for Research in Sleep (SRS) and the Society for Research in Biological Rhythms (SRBR) addressed sleep and circadian rhythms to identify diagnostic and treatment needs for high-priority testing in CRSWD. In these disorders, the duration of the primary sleep episode is earlier or later than expected (it is present in SAD—Seasonal Affective Disorder). The circadian abnormalities exemplified by several adverse health outcomes associated with CRSWD include major depression, substance use, cellular aging, and even job loss and school truancy [2,4].

When melatonin was administered to regulate sleep in the study by Huang et al., serum BDNF levels increased, indicating an association between melatonin levels and the concentration of BDNF, with the authors of the study pointing to a therapeutic effect of BDNF on CRSWD. Therefore, this study may provide a theoretical basis for the melatonin treatment of CRSWD [5].

There is much evidence to suggest that melatonin can be used to prevent and treat cancer [6,7]. As an organic hormone-like compound, it exhibits anti-inflammatory, antioxidant, and oncostatic activities and potential in combating various malignancies. In the gastrointestinal system, melatonin plays an important role through the membrane receptors MT1 and MT2. It can reduce esophageal lesions resulting from mucosal contact with pepsin and bile. Melatonin’s protective mechanisms for esophageal cancer include the inhibition of myosin light chain kinase expression and extracellular signal transduction protein kinase, and its therapeutic effects are summed. Melatonin’s antitumor efficacy is related to its anti-angiogenic effects, immune regulation, and proapoptotic and anti-proliferative functions. The use of melatonin can increase the efficacy of conventional cancer treatments and survival time in patients [8]. Melatonin stimulates pancreatic enzyme secretion by activating the gut/pancreatic reflex and releasing cholecystokinin (CCK). It has a protective effect on the pancreas, preventing the development of acute pancreatitis and pancreatic damage [9].

Melatonin has oncostatic functions in numerous human malignancies. As a differentiating factor in some tumor cells at fi-jological and pharmacological concentrations, it can reduce invasiveness and metastasis by binding MT1 and MT2 receptors, calmodulin, and quinone reductase II, as well as by binding orphan nuclear receptors (ROR/RZR, retinoid orphan receptors/retinoid Z receptors), which are important in the immune system. Melatonin reduces the invasiveness of human cancers, including, but not limited to, prostate, breast, liver, oral, lung, and ovarian cancers. Melatonin’s oncostatic effects are mediated through antiproliferative effects, immune stimulation, cell cycle modulation, apoptosis, autophagy, modulation of oncogene expression, and anti-angiogenic effects [10].

DNA damage can be caused by reactive nitrogen or oxygen species, alkylating agents, depurination, and depyrimidation. DNA repair pathways can neutralize the negative effects of these agents through several mechanisms, of which homologous recombination repair (HRR) and non-homologous end joining (NHEJ) are important [11]. 

Melatonin significantly improved cardiac function and reduced myocardial apoptosis and oxidative damage [12]. In addition to its role in regulating circadian rhythms, melatonin is a known anti-inflammatory and antioxidant agent [13].

Soluble amyloid-β (Aβ) oligomers are thought to play a key role in the pathogenesis of AD (Alzheimer’s disease), and melatonin, through its role in modulating a wide range of signaling pathways, including Notch1, may be a potential therapeutic agent for AD and possibly other age-related neurodegenerative diseases [14]. Moreover, it was shown that the rs10830963 C/G gene melatonin receptor 1B (MTNR1B) polymorphism was associated with an increased risk of type 2 diabetes (T2DM) [15]. Al-Khafaji et al. noticed that diabetic nephropathy affects antioxidant enzyme activity such as catalase, glutathione peroxidase, paraoxonase 1, and glutathione-s-transferase. Similarly, lipid peroxidation is significantly affected compared with healthy controls. The superoxide dismutase enzyme has some exceptions because its activity is elevated. Moreover, the concentration of melatonin is not affected in diabetic patients without nephropathy, but it decreases significantly in diabetic patients with nephropathy when compared with healthy subjects [16].

Abnormal melatonin synthesis in multiple sclerosis (MS) has been observed in association with the different lifestyles of patients, who sometimes had high levels of this neurohormone. The data emphasize the importance of monitoring diurnal changes in melatonin in every MS patient by considering diet and lifestyle to avoid overdosing on melatonin, which is not a panacea in all conditions. Previously, melatonin was thought to have universal positive regulating effects and that it could not be overdosed [17].

The meta-analysis showed that only controlled-release melatonin (and not the immediate-release formulation) reduced systolic blood pressure during sleep by 3.57 mm Hg. Diastolic blood pressure during sleep and wakefulness was also lowered [18].

The evidence for the impact of magnetic fields (MFs) on melatonin production in humans is, to some extent, limited and not always consistent. Some of the inconsistencies can partly be explained by findings suggesting an interaction with light in pineal gland responses to MF, although it is now known that altering magnetic field parameters can also affect the extent to which melatonin secretion is reduced at night [19].

Epidemiological studies have indicated an increased risk of both adult and child leukemia near overhead high-voltage power lines at distances beyond the range of the electric and magnetic fields generated by the lines. The corona ions created in these fields are emitted by the power lines, forming a plume that is carried by the wind away from the lines for distances of up to several hundred meters. The plume generates highly variable disturbances in the atmospheric electric field of tens to hundreds of V/m over a period of seconds or even minutes. They are believed to cause disruption of nocturnal melatonin synthesis and associated diurnal rhythms, resulting in the risk of adverse health effects, including hematopoietic cancers in adults and children [20].

In the research by the authors of this study, there is evidence of differences in melatonin secretion depending on ELF EMF (extra-low-frequency electromagnetic fields) parameters. The circadian rhythm of serum melatonin concentration was estimated in 12 men with low back pain syndrome before and after exposure to an ultra-low-frequency magnetic field (2.9 mT, 40 Hz, rectangular wave, and bipolar). Patients were subjected to the magnetic field for 3 weeks (20 min per day; 5 days per week) in the morning (at 10 am) or late afternoon (at 6 pm). A significant decrease in the nocturnal growth of melatonin was observed regardless of the exposure time. This effect was characteristic of all subjects, although the percentage reduction in melatonin secretion varied between participants [21].

Therefore, in a subsequent study, we checked whether MF with other parameters had the same effect. Seven men suffering from low back pain were included in the study. The patients were subjected to a pulsating electromagnetic field (induction: 25–80 microT; frequency: 200 Hz; complex saw-shaped impulse; bipolar) generated by the Quantronic MRS 2000 apparatus (it is the same as the Viofor JPS M1P1 program) for 3 weeks (5 days a week, twice a day at 08:00 and 13:00, for 8 min) for the whole body. The study was conducted in the spring. Twenty-four-hour serum melatonin profiles were assessed 1 day before MF exposure (baseline) and 1 day and 1 month after the last exposure. No changes in melatonin concentration were observed either after 1 day or 1 month after exposure compared to the baseline value [22].

The current study is a continuation of the above studies. As indicated by these studies, it is assumed that the magnetic field may disturb the nocturnal secretion of melatonin, depending on the type of parameter. Due to the use of magnetic stimulation in physical therapy, it is important to determine the adverse effects and possible therapeutic effects.

The aim of the study was to assess changes in melatonin concentration under the influence of magnetic stimulation in men with low back pain.

Hypothesis: Magnetic stimulation applications with various parameters have a different influence on serum melatonin concentrations in patients with low back pain.

## 2. Results

The obtained results indicate that the applied magnetic stimulation programs did not cause a decrease in the nighttime peak of melatonin concentration.

Prior to the application of magnetic fields, a significant difference was observed between the melatonin concentrations of the circadian curves in samples taken at 08:00, 12:00, 16:00, and 20:00 (Figure 1).

After 15 applications, there was a significant difference between melatonin concentrations only at 08:00 (Figure 2).

Previously occurring differences in melatonin concentrations at 12:00, 16:00, and 20:00 have been cancelled out by approximating the neurohormone curves of the two programs, which was mainly due to a change in the melatonin curve after using the M1P1 program.

One month after the end of the procedures, no significant differences during the daily curves were observed. The changes in concentrations occurred under the influence of the M2P2 program at 08:00 (Figure 3).

## 3. Discussion

There is substantial evidence that exposure to magnetic fields (MFs) affects melatonin secretion in animals. However, data on its effects on melatonin levels in humans are still sparse and appear to be inconclusive at times. Due to the many beneficial effects, very low-frequency MF exposure is used in the physiotherapy of some neurological diseases and overload syndromes of the musculoskeletal system. In previous studies, a decrease in nocturnal melatonin concentrations in human serum was observed after exposure to MFs in magnetic therapy (2.9 mT, 40 Hz, square wave, bipolar), and no decrease as a result of magnetic stimulation with the Quatronic MRS 2000 (exponential shape, bipolar) was confirmed. It has been suggested that differences between various studies may be due to the different characteristics of the MF used. Therefore, in this study, we examined whether magnetic stimulation with parameters different from those mentioned in magnetic stimulation using Viofor JPS exerts a suppressive effect on melatonin secretion or differs from each other in some other way [21,22].

In the case of magnetic environmental and industrial fields, one of the side effects of any electrical device is the electromagnetic field generated near it when operating, which affects the health of employees. All organisms, including humans, are exposed to the impact of various types of environmental fields, characterized by various physical parameters, on a daily basis. Therefore, it is important to precisely determine the influence of the electromagnetic field on physiological and pathological processes occurring in cells, tissues, and organs. Numerous epidemiological and experimental data suggest that extremely low-frequency magnetic fields generated by power lines and devices powered by electricity, as well as high-frequency electromagnetic radiation emitted by electronic devices, have a potentially negative impact on the circadian cycle [23]. Hence, it is important that physical therapy devices do not cause disturbances in the circadian cycle. Such disorders are present in the circadian cycle in children with Attention Deficit Hyperactivity Disorder (ADHD), Parkinson’s disease, or Alzheimer’s disease, as well as in shift workers [3,24,25,26]. The magnetic field is used as a physical treatment in degenerative diseases of the brain; therefore, it is important not to deprive patients of the beneficial effects of melatonin and to not stimulate the secretion of cortisol outside the proper circadian rhythm [27,28].

Evidence of low quality (downgraded due to imprecision) presented by Andersson et al. [29] suggests that 4 to 8 weeks of melatonin treatment reduces pain intensity compared with a placebo. The analgesic effect is confirmed in the work of Ahmad et al. [6]. The use of melatonin can even include situations related to the wide use of melatonin in treating various gastrointestinal diseases, inflammations, cancers, mood disorders, and others. Magnetic fields can also affect the perception of pain [30]. Most of melatonin’s actions are receptor-mediated; however, a few are molecule-mediated, a prominent example being free radical scavenging activity. Melatonin, being a powerful antioxidant, is directly responsible for scavenging free radicals, just like its metabolites. The effect of free radicals and anti-inflammatory has a significant impact on inflammation and the etiology of pain [6].

Melatonin’s various functions as an antioxidant and potential medicine show that it is more than just a sleep hormone synthesized by the pineal gland. Studies have also shown its ability to fight oxidative stress as a free radical scavenger or by stimulating the activation of antioxidant enzymes. Nonionizing electromagnetic fields, although regarded as safe, cause biological changes in the body, creating a condition of excessive oxidative stress and intensifying lipid peroxidation. Some studies on the effects of electromagnetic fields on melatonin in humans have provided conflicting evidence, while others have reported only negative results. However, the possibility of electromagnetic field impact on the usefulness of melatonin in the body to control external stressors has oriented research toward investigating the radio-protective role of melatonin [31].

Human and rat results are similar when the MF strength is limited to one range (with B ≲ 50 μT). In addition, the data reveal that chronic exposure (greater than ~22 days) to ELF-MF appears to decrease melatonin levels only when MF intensity is below the ~30 μT threshold, i.e., when man-made ELF-MF intensity is below the static geomagnetic fields [32].

The modern human population is widely exposed to magnetic fields (50 Hz in Europe and 60 Hz in North America), which are produced by various electrical devices commonly used in homes and workplaces. Reduced urinary excretion of 6-sulfatoxymelatonin has been observed in many studies in electrical plant workers who were exposed to 60 Hz magnetic fields [23]. Significant changes were seen after the second day of the work week, with the effect of magnetic field exposure being most pronounced in those with low exposure to light in the workplace. This is a slightly different observation than in the studies of Juutilainen et al., because in this case, a greater decrease in melatonin concentration was observed in people more exposed to light [19]. Nevertheless, it is clear that environmental and network fields interfere with nocturnal melatonin secretion, with subsequent negative health effects [23].

In the studies analyzed in the above-quoted paper, parameters of the magnetic field other than strength were not taken into account. Magnetic stimulation was applied using a Viofor JPS device with a mat, creating a sawtooth-edge-shaped basal pulse (exponential). The P2 Program using a resonant cyclotron mechanism with 0–300 µT induction was configured to the M2 application mode with growing induction for 12 min.

The Viofor JPS Classic, Viofor JPS Clinic, and Viofor JPS Delux signal patterns are set in three programs: P1, P2, and P3. The frequencies of the basic pulses are in the range 180–195 Hz. The frequencies of the pulse packs are in the range of 12.5–29 Hz, groups 2.8–7.6 Hz, and series 0.08–0.3 Hz. The individual pulses assume a complex ball-shaped (exponential) bipolar shape. In its ascending section, M2P2, there are two parts of a linearly increasing induction with a variable inclination, which is intersected by a constant induction (pulse type I M2P2 and II M3P3 (Figure 4)).

It cannot be ruled out that the change in the frequency sequence and waveform, as well as the presence of cyclotron resonance, determined the changes in the melatonin curve in the 24 h cycle after using the M2P2 program, as can be observed a month after the end of the applications [33]. The fact that magnetic stimulation does not reduce the nocturnal peak of melatonin secretion suggests that it is effective in the analgesic effect in the presence of other analgesic mechanisms provided by magnetic fields.

The results shown in the present study indicate the effect of the M1P1 program as normalizing melatonin concentrations in the midday hours, observed one day after the end of application. In both methods of magnetic field exposure (M1P1 and M2P2 program), the timing of melatonin secretion is prolonged in the morning hours one month after the end of treatment, which may have implications for patients with SAD, who tend to have difficulty getting up in the morning [2,34]. This can have a negative effect on the mood and pain of people treated for low back pain. It is evident, however, that magnetostimulation does not work like net fields. Furthermore, it regulates the physiological curve of melatonin secretion. The type of selected parameters, the system of applications, and the time and duration of applications have a major impact here, which is visible in other charts of the melatonin circadian cycle after magnetic stimulation in two programs. Another effect of magnetic stimulation after 1 month from the end of the application was the intensification of the previously obtained results, which confirms the hypothesis about the appearance of the phenomenon of biological hysteresis. Magnetic stimulation seems to have unifying properties that regulate the secretion of melatonin.

Interestingly, despite the increase in serotonin concentration in the M2P2 program, it was not reflected in the circadian curve of melatonin [35], which would be related to the common metabolic chain of serotonin-to-melatonin conversion. It is likely that the effect on the conversion of serotonin to melatonin does not depend on the stimulation of serotonin N-acetyltransferase from acetyl-CoA or 5-hydroxyindole-O-methyltransferase. Magnetic stimulation of M2P2 may affect the synthesis of serotonin without changing the intensity of the transformation into melatonin. 

The obtained results indicate that magnetic stimulation has an impact on melatonin levels in men with low back pain, although it works at different times depending on the programs and the application. However, it requires further analyses in a larger group of patients in correlation with the level of pain and supplemented with long-term follow-up. It would be worth comparing the possible impact of the M3P3 Viofor JPS program (Figure 4) to those tested here with respect to the different parameters of the basic signal. However, as it seems, knowing all the mechanisms that determine the action of melatonin is still in the area of further research [36].

The study is limited to small groups of patients, which is due to the difficulty of hospitalizing a larger group of men in the ward at the same time to maintain a similar length of days and nights in the diurnal cycle. Another problem was maintaining for a larger group of the others the same conditions of daily rhythm (e.g., meals, exercise) and uniform organization of the study. Furthermore, the study was not sponsored and was performed at the expense of the investigators. However, in other human and animal studies, the groups were small, and melatonin was taken only once a day [37,38], probably for the reasons mentioned above. Despite the large amount of research related to the diurnal cycle of humans and animals under the influence of environmental pollution by magnetic fields, further research on melatonin secretion seems necessary since, as research indicates, even small differences in the parameters of magnetic fields cause a variety of responses of organisms in the secretion of melatonin in the diurnal cycle.

## 4. Materials and Methods

### 4.1. Study Population

The study was approved by the Bioethics Committee for Scientific Research at Medical University in Lodz, number RNN/254/05/KB.

The study was conducted on 15 men with low back pain syndrome. The patients were treated in the Rehabilitation Department of the Independent Public Healthcare Complex in Sieradz, Poland.

Inclusion criteria are as follows:
-Men aged 18 to 60 years with chronic low back pain;-The men did not suffer from any chronic or acute diseases of the digestive system, circulatory system, or metabolism and had no hormonal disorders;-Not taking any medications;-No contraindications to physiotherapy from any systems or organs;-Consent to examination procedures (seven-week hospital stay; staying 8 h a day at night with dim red light; having blood drawn 8 times a day, every 2 h at night; following a daily routine of meals and treatments).

Exclusion criteria are as follows:
-Female sex (avoiding the impact of melatonin fluctuations related to the menstrual cycle);-Contraindications to the use of a magnetic field;-The intensity of pain was no more than 3 degrees on the 10-degree VAS scale;-Lack of consent from the patient and the caregiver for research and participation in the program.

Only men were included in the study due to the fact that melatonin concentrations change in women during the menstrual cycle, and the study group would not be homogeneous in terms of hormonal regulation.

The patients were divided into two groups. Group 1 consisted of 7 men with an average age of 37 years (32–42), who were administered the M1P1 program of the Viofor JPS device without cyclotron resonance twice a day, for 8 min, at 08:00 and 13:00. The entire body of the patient lying on a mat-shaped applicator was exposed to the magnetic field. Group 2 consisted of 8 men with an average age of 46 years (38–57), who were administered the M2P2 program of the Viofor JPS device once a day for 12 min at 10:00. The entire body of the patient lying on a mat-shaped applicator was exposed to the magnetic field, as in Group 1. The subjects were exposed to an illumination rhythm of 8 h per day in red light and 16 h per day in artificial light, corresponding in intensity to daily natural light. Melatonin was collected according to the Journal of Pineal Research guidelines for authors [39]. Thus, in general, they were not subjected to the natural daily changes in illumination associated with the seasons. The shape of a single pulse for M2P2 and M3P3 is shown in Figure 4 below. The pulse shape in the M1P1 program is without horizontal inserts of constant induction and does not have the cyclotron resonance feature.

### 4.2. Test Protocol for Magnetic Fields

Low-frequency magnetic fields used in physiotherapy have only non-thermal effects such as improving tissue oxygen utilization, anti-inflammatory, anti-oedema, angiogenetic, vasodilator, pro-cognitive, and tranquilizing effects. They enhance healing wounds and bone fractures; they reduce blood pressure; and they improve blood rheological parameters. Magnetic fields modify blood glucose and cholesterol levels. The analgesic effect of magnetic fields depends on the parameters used. In the cited study, the use of magnetic stimulation (M1P1, M2P2, and M3P3 programs) was more effective than the use of magnetotherapy with parameters of 2.9 mT, 40 Hz, rectangular wave, 20 min. The results of all groups were significant relative to placebo (shame exposition). As a result of magnetic stimulation, a biological hysteresis effect is noted, i.e., the effect of the magnetic field remaining in place for a month after the application [21,23,26].

The main difference between the two treatments is the induction and frequency of the applied magnetic field, as well as the waveform. Magnetic stimulation uses much lower induction levels and higher field frequencies, up to 3000 Hz. However, the envelopes of the transmitted signals at lower frequencies, even a few Hz, generating ion cyclotron resonance, have a significant effect. The signal structure of Viofor JPS Classic, Viofor JPS Clinic, and Viofor JPS Delux is determined by three programs: P1, P2, and P3. The fundamental pulse frequencies are contained within the range 180–195 Hz. Pulse pack frequencies are in the range 12.5–29 Hz, groups 2.8–7.6 Hz, and series 0.08–0.3 Hz. The individual pulses assume a complex shape resembling a sawtooth (exponential). In its ascending section, there is a part of a linearly increasing induction that is crossed by a constant induction (pulse types I and II). Large mat-like applicators with uniform field induction were used in the investigations [36,40].

Magnetic fields were applied to each group for 3 weeks, 5 days a week, with a weekend break. The daily melatonin profile was estimated the day before exposure to magnetic stimulation (as control) and the day after 15 applications. Additionally, melatonin concentrations were estimated in both groups one month after all applications. Each subject served as his own control.

### 4.3. Study Protocol—Blood Analysis

Blood samples were taken in both groups at 08:00, 12:00, 16:00, 20:00, 00:00, 02:00, 04:00, and 08:00. The times reflect well the diurnal rhythm of melatonin secretion and exclude dietary effects on serum hormone concentrations. Meal times were 09:00, 13:00, and 17:00. At night, samples were taken under dim red light. Patients were in such light from 22:00 to 06:00 the day before and on the night of sampling. During the day, the subjects were exposed to artificial light similar in intensity and color to daylight for 16 h. The tests in group 1 were carried out in early spring and in group 2 in late autumn. Seasonality had no effect on melatonin concentrations due to the form of lighting [30].

The serum received by centrifugation was stored at −20 °C until melatonin concentrations were determined. Concentrations were assessed using an RIA kit (DRG instr. GmbH, Marburg, Germany, cat. no. IH RH 29301), intra-test CV 8%, and inter-test CV 14.8%.

### 4.4. Data Analysis 

In attempt to answer the research questions set, statistical analyses were undertaken using STATISTICA StatSoft Poland Version 8.

Statistical analysis was done using the Student’s *t*-test for matched pairs and the Wilcoxon signed rank test with a significance level of *p* ≤ 0.05.

## 5. Conclusions

The studies show the effect of the M1P1 program on normalizing the concentration of melatonin in the afternoon one day after applications. In both exposure methods, the duration of melatonin secretion is prolonged in the morning hours one month after applications, which may affect the condition of patients with SAD.

The effect of the magnetic field was maintained after 1 month from the end of the application, which confirms the hypothesis about the occurrence of the phenomenon of biological hysteresis after the use of magnetic stimulation.

The parameters of magnetic fields, the system of applications, and the time and duration of applications may influence the secretion of melatonin, which is visible in other melatonin circadian cycle charts after magnetic stimulation in two programs. The impact of such parameters as the type of polarization and the vector should be taken into consideration.

Magnetic stimulation seems to have the properties of a unifying nature that regulate the secretion of melatonin, which, however, requires further study.

## Figures and Tables

**Figure 1 ijms-24-15860-f001:**
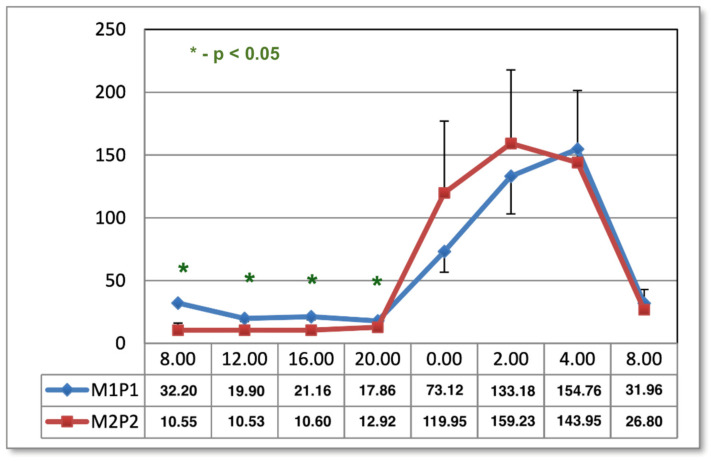
The circadian concentrations (ng/mL) of melatonin before applications of magnetic stimulation. *—statistical significance.

**Figure 2 ijms-24-15860-f002:**
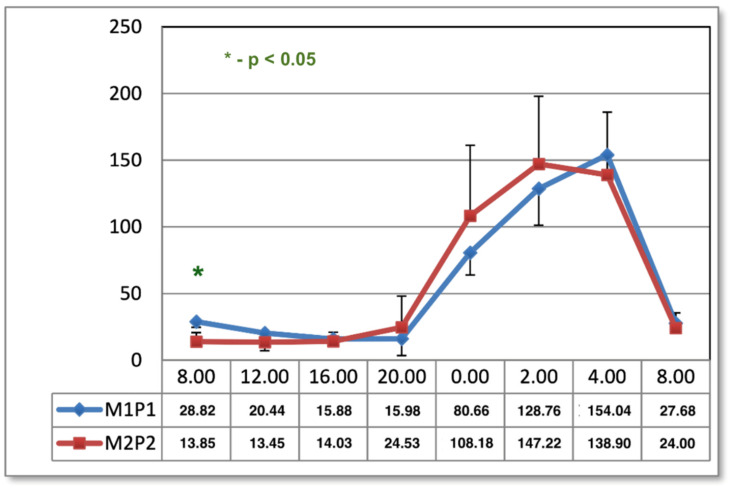
The circadian concentrations (ng/mL) of melatonin after 15 applications of magnetic stimulation. *—statistical significance.

**Figure 3 ijms-24-15860-f003:**
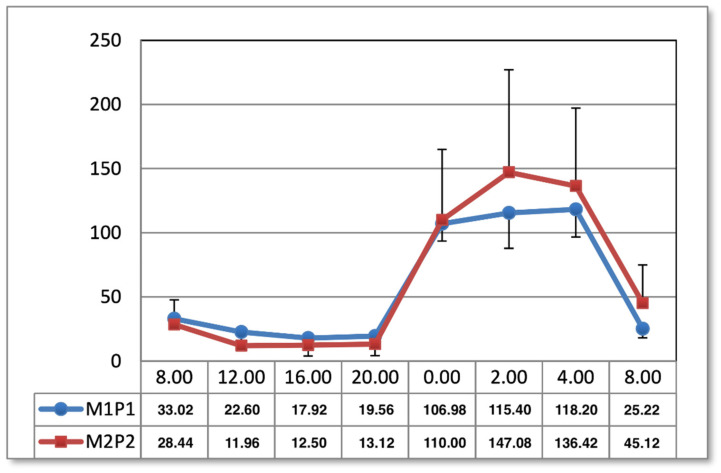
Comparison of the circadian concentration (ng/mL) of melatonin one month after the end of application magnetic stimulation.

**Figure 4 ijms-24-15860-f004:**
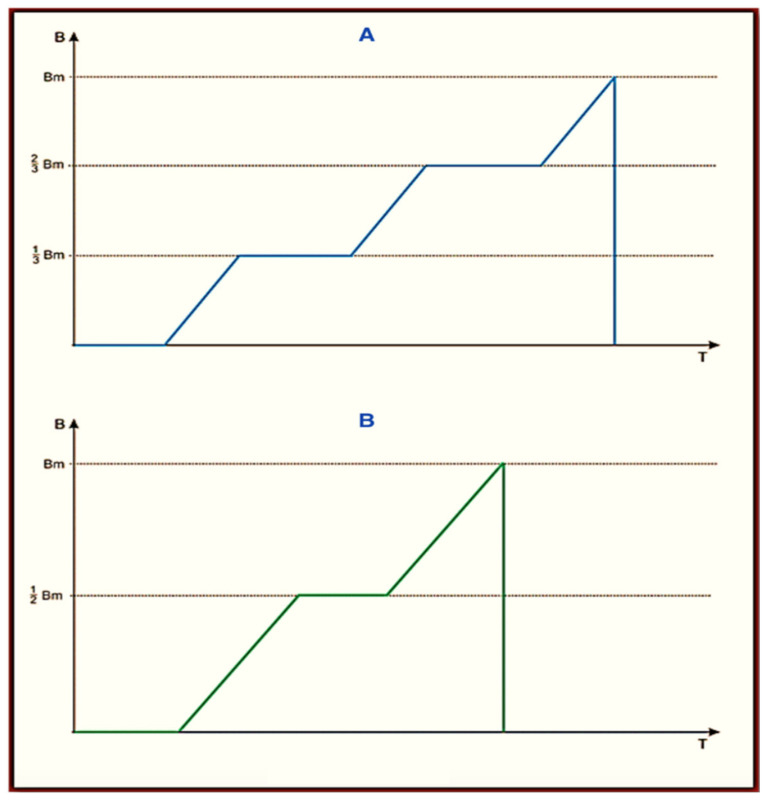
Shapes of single impulses generated by Viofor JPS having features of magnetic resonance in programs M2P2 type I (**A**) and M3P3 type II (**B**).

## Data Availability

The data are available from the corresponding author if required.

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
