# Peer review of "Changes in Melatonin Concentration in a Clinical Observation Study under the Influence of Low-Frequency Magnetic Fields (Magnetic Stimulation in Men with Low Back Pain)—Results of Changes in an Eight-Point Circadian Profile"

_ijms, 2023, doi:10.3390/ijms242115860_

Round 1
Reviewer 1 Report
Comments and Suggestions for Authors
Congratulations on writing this article.
Herein, I suggest some amendments for your paper, sincerely.
The introduction section contains informative data; However, some of these data are not relative to the title of the article. I suggest you to add some data and background about the magnetic field exposure and also the relationship between melatonin and patients' condition besides the data provided in the discussion part.
For more data on the roles of melatonin, I recommend using these articles:
1- Melatonin: An anticancer molecule in esophageal squamous cell carcinoma: A mechanistic review
2- Oncostatic activities of melatonin: Roles in cell cycle, apoptosis, and autophagy
3- Melatonin: A smart molecule in the DNA repair system
4- The relationship between melatonin level and antioxidant enzymes in diabetic patients with and without nephropathy
Dear colleagues, the Materials and Method section was written well however it should be written after the introduction. Also explaining M1P1 and M2P2 program in details would be nice.
The data presented in result part was clear and this part was written adequately.
As for the discussion section, the results were properly described.
Author Response
Reviewer one
Congratulations on writing this article.
Herein, I suggest some amendments for your paper, sincerely.
- The introduction section contains informative data; However, some of these data are not relative to the title of the article. I suggest you add some data and background about the magnetic field exposure and also the relationship between melatonin and patients' condition besides the data provided in the discussion part.
Dear Reviever,
Thank you very much for your insightful and kind review. The following articles were found by us, and their contents to the extent of completeness were included in the introduction of the article.
Melatonin: An anticancer molecule in esophageal squamous cell carcinoma: A mechanistic review
Heydari, N., Memar, M.Y., Reiter, R.J., Rezatabar, S., Arab-Bafran, Z. i, Jaz, A.A. and Mir, S.M. 2023. Melatonin: An anticancer molecule in esophageal squamous cell carcinoma: A mechanistic review. Melatonin Research. 6, 1 (Feb. 2023), 59-71. DOI:https://doi.org/https://doi.org/10.32794/mr112500141.
Oncostatic activities of melatonin: Roles in cell cycle, apoptosis, and autophagy
Targhazeh N, Reiter RJ, Rahimi M, Qujeq D, Yousefi T, Shahavi MH, Mir SM. Oncostatic activities of melatonin: Roles in cell cycle, apoptosis, and autophagy. Biochimie. 2022 Nov;202:34-48. doi: 10.1016/j.biochi.2022.06.008. Epub 2022 Jun 23. PMID: 35752221.
Melatonin: A smart molecule in the DNA repair system
Mir SM, Aliarab A, Goodarzi G, Shirzad M, Jafari SM, Qujeq D, Samavarchi Tehrani S, Asadi J. Melatonin: A smart molecule in the DNA repair system. Cell Biochem Funct. 2022 Jan;40(1):4-16. doi: 10.1002/cbf.3672. Epub 2021 Oct 21. PMID: 34672014.
The relationship between melatonin level and antioxidant enzymes in diabetic patients with and without nephropathy
Al-Khafaji, A., Mir, S., Mohammadzadeh, F., Abolghasemi, M., & Hadwan, M. (2022). The relationship between melatonin level and antioxidant enzymes in diabetic patients with and without nephropathy. Baghdad Journal of Biochemistry and Applied Biological Sciences, 4(02). https://doi.org/10.47419/bjbabs.v4i02.207
- Dear colleagues, the Materials and Method section was written well however it should be written after the introduction. Also explaining M1P1 and M2P2 program in details would be nice.
Dear Reviver,
The article for the journal is sent in the form of a template. The order of subsections is determined by the publisher.
In the aspect of the description of the mentioned programs, we refer to an earlier article.
Woldańska-Okońska M, Czernicki J, Karasek M. The influence of the low-frequency magnetic fields of different parameters on the secretion of cortisol in men. Int J Occup Med Environ Health. 2013 Mar;26(1):92-101. doi: 10.2478/s13382-013-0090-6. Epub 2013 Apr 12. PMID: 23576151.
In which both magnetostimulation application programs are antedated, with a focus on the shape of the pulse, as well as how they are applied.
The authors were concerned about self-plagiarism, as the article cited in the current paper also presented the characteristics of magnetic fields.
WoldaÅ„ska-OkoÅ„ska, M.; Koszela, K. Chronic-Exposure Low-Frequency Magnetic Fields (Magnetotherapy and Magnetic Stimu-lation) Influence Serum Serotonin Concentrations in Patients with Low Back Pain—Clinical Observation Study. Int J Environ Res Public Health 2022, 19, 9743. https://doi.org/10.3390/ijerph19159743
- The data presented in result part was clear and this part was written adequately.
Thank you
- As for the discussion section, the results were properly described.
Thank you
Reviewer 2 Report
Comments and Suggestions for Authors
The study titled "Changes in Melatonin Concentration in Clinical Observation Study Under the Influence of Low Frequency Magnetic Fields (Magnetic Stimulation in Men with Low Back Pain) - Results of Changes in Eight Point Circadian Profile" aims to investigate the effects of magnetic stimulation on melatonin secretion in individuals suffering from low back pain.
The authors hypothesize that different parameters of magnetic stimulation might have varying effects on serum melatonin concentrations. While the study provides some interesting insights into the influence of magnetic fields on melatonin, it has several serious limitations and definitely lacks clinical data correlations.
One of the primary limitations of this study is the absence of clinical relevant data on the patients group. The study assesses the changes in melatonin concentration under the influence of magnetic stimulation, but considering the vast implications of melatonin in the circadian and redox homeostasis of the human organism, the lack of any relevant clinical data is the major drawback I identified. The authors provide no details on the circadian profile of the patients, the associated pathologies, the eventual administration of drugs due to the chronic profile of their pain, the correlations with the age of the patients, as we know that the pineal secretion decreases with age, a coherent sleeping profile of the subjects. To establish the clinical relevance of the study's findings, it is essential to correlate melatonin changes with meaningful clinical endpoints.
Considering the small sample size, the authors acknowledge that they faced difficulties in recruiting a larger group of participants. However, small sample sizes can lead to unreliable results and may not accurately represent the broader population. To enhance the study's credibility and generalizability, it is imperative to conduct further investigations with larger and more diverse groups of patients, and to provide the complete details regarding their clinical profile and lifestyle references.
The study recognizes the complexity of melatonin's action and suggests that there is more to learn about the mechanisms involved. To enhance the study's value, further serious research should explore additional parameters and mechanisms that influence melatonin secretion, providing a more comprehensive understanding of the subject matter.
In conclusion, the study offers valuable insights into the impact of magnetic stimulation on melatonin levels in individuals with low back pain. However, the research is limited by its small sample size, lack of clinical data correlations, and incomplete follow-up, to a longer period of time (over 6 months). To strengthen the study's contribution to the field, the authors should consider conducting future research with larger sample sizes, conducting thorough clinical correlations, and providing a more in-depth follow-up analysis. Addressing these issues will help ensure the study's findings are both scientifically rigorous and clinically meaningful.
Comments on the Quality of English Language
No major concerns regarding English language.
Author Response
The study titled "Changes in Melatonin Concentration in Clinical Observation Study Under the Influence of Low Frequency Magnetic Fields (Magnetic Stimulation in Men with Low Back Pain) - Results of Changes in Eight Point Circadian Profile" aims to investigate the effects of magnetic stimulation on melatonin secretion in individuals suffering from low back pain.
The authors hypothesize that different parameters of magnetic stimulation might have varying effects on serum melatonin concentrations. While the study provides some interesting insights into the influence of magnetic fields on melatonin, it has several serious limitations and definitely lacks clinical data correlations.
Dear Reviever,
Thank you very much for your insightful and detailed review.
- One of the primary limitations of this study is the absence of clinical relevant data on the patients group. The study assesses the changes in melatonin concentration under the influence of magnetic stimulation, but considering the vast implications of melatonin in the circadian and redox homeostasis of the human organism, the lack of any relevant clinical data is the major drawback I identified. The authors provide no details on the circadian profile of the patients, the associated pathologies, the eventual administration of drugs due to the chronic profile of their pain, the correlations with the age of the patients, as we know that the pineal secretion decreases with age, a coherent sleeping profile of the subjects. To establish the clinical relevance of the study's findings, it is essential to correlate melatonin changes with meaningful clinical endpoints.
The study included the following inclusion criteria:
– men aged 18 to 60 years with chronic low back pain;
– the men did not suffer from any chronic and acute diseases of the digestive system, circulatory system, or metabolism, and had no hormonal disorders;
– not taking any medications;
– no contraindications to physiotherapy from any systems or organs;
– consent to examination procedures (seven-week hospital stay, staying 8 hours a day at night with dim red light, having blood drawn 8 times a day - every 2 hours at night, following a daily routine of meals and treatments).
The study included the following exclusion criteria:
– female sex (avoiding the impact of melatonin fluctuations related to the menstrual cycle)
– contraindications to the use of a magnetic field,
–the intensity of pain was no more than 3 degrees on a 10-degree VAS scale,
– lack of consent from the patient and the caregiver for research and participation in the programme.
The subjects were exposed to an illumination rhythm of 8 hours per day in red light and 16 hours per day in artificial light, corresponding in intensity to daily natural light. Melatonin was collected according to the Journal of Pineal Research guideline for authors. Thus, in general, they were not subjected to the natural daily changes in illumination as-sociated with the seasons.
In addition, patients had a unified diet, and meal times did not affect melatonin secretion, as they were given 3 times a day after sampling. The daily exercise cycle was the same for the subjects, as well as the time of magnetic stimulation application. In our earlier study, there was no correlation between magnetic field application times and the diurnal curve of melatonin.
Karasek, M.; Woldanska-Okonska, M.; Czernicki, J.; Zylinska, K.; Swietoslawski, J. Chronic exposure to 2.9 mT, 40 Hz magnetic field reduces melatonin concentrations in humans. J Pineal Res 1998, Dec;25(4):240-4. doi: 10.1111/j.1600-079x.1998.tb00393.x.
Generally, there are no studies in the literature on melatonin concentrations after half a year after field application. There are rare results reported after a few dozen days.
- Considering the small sample size, the authors acknowledge that they faced difficulties in recruiting a larger group of participants. However, small sample sizes can lead to unreliable results and may not accurately represent the broader population. To enhance the study's credibility and generalizability, it is imperative to conduct further investigations with larger and more diverse groups of patients, and to provide the complete details regarding their clinical profile and lifestyle references.
A larger group of patients would be difficult to organize in terms of maintaining identical diurnal cycle conditions, including light exposure, technical capabilities for blood collection, exercise application and magnetic stimulation. It should be remembered that this was a hospitalized group, and usually in rehabilitation patients are characterized by multi-disease. In contrast, our patients were healthy men with no comorbidities and were not taking any medications.
The intensity of pain was negligible in them (up to 3 degrees on a 10-degree VAS scale) and was, so to speak, a pretext for hospitalization, which in the therapeutic aspect was primarily intended to have a prophylactic effect (ergonomics of daily activities, teaching exercise)
Most of the studies of melatonin concentrations on a daily basis, are small groups of people, because collecting blood 8 times a day is not only a logistical challenge, but also a general human one. E.g., patients were awakened every 2 hours during the night, and blood was drawn in dim red light.
To illustrate these circumstances, a study in animals can be cited:t:Kazemi M, Sahraei H, Aliyari H, Tekieh E, Saberi M, Tavacoli H, Meftahi GH, Ghanaati H, Salehi M, Hajnasrollah M. Effects of the Extremely Low Frequency Electromagnetic Fields on NMDA-Receptor Gene Expression and Visual Working Memory in Male Rhesus Macaques. Basic Clin Neurosci. 2018 May-Jun;9(3):167-176. doi: 10.29252/NIRP.BCN.9.3.167. PMID: 30034647; PMCID: PMC6037432,
In which four monkeys were studied.
Similarly, in humans:
Wu, X.; Bai, F.; Wang, Y.; Zhang, L.; Liu, L.; Chen, Y.; Li, H.; Zhang, T. Circadian Rhythm Disorders and Corresponding Function-al Brain Abnormalities in Young Female Nurses: A Preliminary Study. Front Neurol 2021 Apr 30;12:664610. doi: 10.3389/fneur.2021.664610. PMID: 33995261; PMCID: PMC8120025.
Nine nurses participated in the study, and measurements were taken 5 times per daily cycle. This was a one-time measurement, and there was no subsequent follow-up.
- The study recognizes the complexity of melatonin's action and suggests that there is more to learn about the mechanisms involved. To enhance the study's value, further serious research should explore additional parameters and mechanisms that influence melatonin secretion, providing a more comprehensive understanding of the subject matter.
The study was concerned with establishing a relationship between magnetic stimulation with specific parameters and melatonin concentration. The essence was the conclusions suggesting that the concentration of melatonin in the diurnal cycle depends on the parameters of the field, which was already demonstrated early on in previous studies. The mechanisms of the field's action are already partially known, and are based on the activity of free radical pairs and the action of nitric oxide, which of course has further molecular consequences.
Jammoul M, Lawand N. Melatonin: a Potential Shield against Electromagnetic Waves. Curr Neuropharmacol. 2022 Mar 4;20(3):648-660. doi: 10.2174/1570159X19666210609163946. PMID: 34635042; PMCID: PMC9608227.
Jammoul M, Lawand N. Melatonin: a Potential Shield against Electromagnetic Waves. Curr Neuropharmacol. 2022 Mar 4;20(3):648-660. doi: 10.2174/1570159X19666210609163946. PMID: 34635042; PMCID: PMC9608227.
Undoubtedly, understanding the mechanisms of melatonin secretion requires further research. On the other hand, interesting in the presented study is the course of the curve after magnetic field application, which has clinical implications. Like the development of long-term changes in the curve.
- In conclusion, the study offers valuable insights into the impact of magnetic stimulation on melatonin levels in individuals with low back pain. However, the research is limited by its small sample size, lack of clinical data correlations, and incomplete follow-up, to a longer period of time (over 6 months). To strengthen the study's contribution to the field, the authors should consider conducting future research with larger sample sizes, conducting thorough clinical correlations, and providing a more in-depth follow-up analysis. Addressing these issues will help ensure the study's findings are both scientifically rigorous and clinically meaningful.
All comments were included in the limitations of the study, as well as a statement that further research is required. This was also raised in the conclusions.
In addition, the English language was proofread.
We therefore ask you to take the above clarifications into account and accept the article for publication.
Round 2
Reviewer 2 Report
Comments and Suggestions for Authors
I do accept the article in the present form for publication, despite the limitations of the study and still important unaddressed issues revealed by my previous comments, as this study could be the starting point of other more detailed experiments. I also appreciate the hard work of the nurses and authors and difficult conditions implied by melatonin collection 5 times a day, in dim light settings, fully respecting the subjects implicated in this study.